# Harnessing *Paenarthrobacter ureafaciens* YL1 and *Pseudomonas koreensis* YL2 Interactions to Improve Degradation of Sulfamethoxazole

**DOI:** 10.3390/microorganisms10030648

**Published:** 2022-03-18

**Authors:** Lan Yu, Yingning Wang, Xiaoqing Shan, Fang Ma, Haijuan Guo

**Affiliations:** 1State Key Laboratory of Urban Water Resource and Environment, School of Environment, Harbin Institute of Technology, Harbin 150090, China; yulan19920221@163.com (L.Y.); lina_wang2@163.com (Y.W.); 21b929035@stu.hit.edu.cn (X.S.); 2College of Energy and Environmental Engineering, Hebei University of Engineering, Handan 056038, China

**Keywords:** sulfamethoxazole, synergistic degradation, consortium, *Pseudomonas koreensis*

## Abstract

Sulfamethoxazole (SMX) is a widespread and persistent pollutant in the environment. Although the screening and analysis of SMX-degrading bacteria have been documented, the interaction mechanisms of functional microorganisms are still poorly understood. This study constructed a consortium with strain YL1 and YL2 supplied with SMX as the sole carbon and energy source. The coexisting mechanism and the removal of SMX of the consortium were investigated. The total oxidizable carbon (TOC) removal rate of the combined bacterial system was 38.94% compared to 29.45% for the single bacterial system at the same biomass. The mixed bacterial consortium was able to resist SMX at concentrations up to 400 mg/L and maintained a stable microbial structure at different culture conditions. The optimum conditions found for SMX degradation were 30 °C, pH 7.0, a shaking speed of 160 r·min^−1^, and an initial SMX concentration of 200 mg·L^−1^. The degradation of SMX was accelerated by the addition of YL2 for its ability to metabolize the key intermediate, 4-aminophenol. The removal rate of 4-aminophenol by strain YL2 reached 19.54% after 5 days. Genome analysis revealed that adding riboflavin and enhancing the reducing capacity might contribute to the degradation of SMX. These results indicated that it is important for the bioremediation of antibiotic-contaminated aquatic systems to understand the metabolism of bacterial communities.

## 1. Introduction

Sulfamethoxazole (SMX) is one of sulfonamide antibiotics (SNAs), which were the first antibiotics systematically used as growth promoters for human and animal drugs due to their antimicrobial properties [1,2]. The ongoing COVID-19 has led to a significant increase in the demand of antibiotics [3]. However, the inefficient treatment toward SNAs by WWTPs has resulted in their widespread distribution in the environment as hazardous organic pollutants [4,5,6,7,8]. SMX was reported to have high concentrations around 4870 ng·L^−1^ in river of China [9]. Currently, industrial wastewater associated with SMX production and wastewater resulting from excessive use of SMX require safe treatment.

Biological treatment uses microbial interaction to remove SMX, which is currently the most promising processing method [10]. To overcome the persistent toxicity and the high environmental mobility of SMX, various removal technologies have been reported, including biological treatment [11,12,13,14], adsorption [15,16], advanced oxidation [17,18,19,20], and electrochemical oxidation [21,22]. By comparison, biological treatment technologies have been widely used for their low treatment cost and environmental friendliness, which mostly rely on the metabolism of microbes [23]. Due to its complex chemical composition, SMX is difficult to be removed directly by microorganisms [24]. In addition, SMX inhibits bacterial folate synthesis, which may make the application of conventional treatment technologies in wastewater treatment plants (WWTPs) inefficient [14]. To eliminate SMX, the presence of degrading bacterial species in WWTPs is crucial [25].

The degradation of SMX by both isolated and enriched bacteria has been reported [26,27,28,29,30,31,32]. Co-cultures are superior to monocultures in biodegradation of SMX, because microorganisms can cooperate with each other to remove antibiotics [33,34]. Sulfonamides-degrading microbial populations have been evaluated by 16S rRNA-based molecular methods in some previous studies, however, the simple focus on the phylogenetic classification of the consortiums cannot lead to the fully understand their functions [35,36]. Studies on the mechanisms of SMX biodegradation by microbial communities can help control and utilize the consortiums, but so far, little is known about the relationships among two or more bacteria in the SMX biodegradation process, and the methodology used to analyze these relationships is also rare.

In this study, the mechanism of SMX degradation by a consortium consist of strain YL1 and strain YL2 was investigated. Strain YL1 was previously isolated from the aeration tank in a municipal wastewater treatment plant, and identified as *Paenarthrobacter ureafaciens* (CGMCC 18365) [25]. Strain YL2 was a novel strain isolated in this study, which could facilitate the SMX degradation when co-cultured with YL1. In the presence of YL1, the sulfonamide bond (S-N) in the SMX molecule was broken, and the benzene ring part was further transformed while the heterocyclic moiety accumulated as a dead-end product [25,37]. Strain YL2 could further metabolize 4-aminophenol completely, which reduced the toxic burden of metabolites and further accelerated the initial degradation of SMX.

## 2. Materials and Methods

### 2.1. Chemicals and Media

SMX and 4-aminophenol (4-AP) were purchased from Tokyo Chemical Industry. 3-amino-5-methylisoxazole (3A5MI) was supplied by Kmaels. Acetonitrile (high-performance liquid chromatography (HPLC) grade) and formic acid (HPLC grade) were purchased from Dikma Technologies Inc., Foothill Ranch, CA, USA. All other chemicals used in this study were analytical grade and obtained from Kermel Chemical Reagent Ltd., Tianjin, China.

The mineral salt medium (MSM) was composed of 0.5 g/L (NH_4_)_2_SO_4_, 1.5 g/L KH_2_PO_4_, 3.5 g/L K_2_HPO_4_, 0.15 g/L MgSO_4_·7H_2_O, 0.5 g/L NaCl and 1.0 mL/L trace elements. In addition, the trace elements used in this study contained NaHCO_3_·10H_2_O 2.0 g/L, MnSO_4_·4H_2_O 0.3 g/L, ZnSO_4_·7H_2_O 0.2 g/L, (NH_4_)_6_Mo_7_O_2_·4H_2_O 0.02 g/L, CuSO_4_·5H_2_O 0.1 g/L, CoCl_2_·6H_2_O 0.5 g/L, CaCl_2_·2H_2_O 0.05 g/L, FeSO_4_·7H_2_O 0.5 g/L. Luria-Bertani medium (LB) was also used: Tryptone 10 g/L, Yeast extract 5 g/L, NaCl 10 g/L. All media were adjusted to 7.0 ± 0.2 using NaOH/HCl. For solid media, 15 g agar was added per liter. All media were sterilized at 120 °C for 20 min before use.

### 2.2. Pure Strain Isolation and Culture Conditions

Enrichment cultures were isolated using a mineral salt medium with 100 mg·L^−1^ SMX as the sole carbon source (SMX-MSM). The consortium colonies were re-purified onto SMX-MSM medium agar three times. We used both SMX-MSM agar medium and LB agar medium to obtain purified isolates.

One SMX-degrading bacterium (YL1) was isolated due to its ability to efficiently metabolize SMX and survive in SMX-MSM at 30 °C and 160 r·min^−1^. When the incubation time was extended to 10 days, distinct different morphotype colonies (orange colonies) were visible around strain YL1 (beige colonies). LB medium was used to isolate the orange colonies that could not survive on SMX-MSM alone and were named as YL2. In the presence of strain YL1, strain YL2 could grow in SMX-MSM. Moreover, Strain YL2 was inoculated into liquid LB medium at 30 °C on an incubator shaker at 160 r·min^−1^.

### 2.3. Identification of Isolates and Morphology Characterization

The isolates were identified by 16S rRNA gene sequence analysis. The sequence of the 16S rRNA gene was determined by Sangon Biotech Co., Ltd. (Shanghai, China). The 16S rRNA gene of the isolated strain was amplified using the universal primers 27F (5′-CAGAGTTTGATCCTGGCTAGGAGGTGATCCAGCCGCA-3′) and 1492R (5′-AGTTTGATCMTGG CTCAGGGTTACCTTGTTACGACTT-3′) and sequenced [25]. The sequence data of the closest relatives were retrieved from the NCBI GenBank and compared with the sequences in the GenBank database using the BLAST program [38]. YL2 was identified by physical-biochemical characterization and 16S rRNA gene sequencing. In addition, strain YL2 (CGMCC No. 18366) was 99% similar to that of *Pseudomonas koreensis* Ps 9-14. The sequences of the 16S rRNA gene from *Pseudomonas koreensis* YL2 were deposited in the GenBank database (accession number MN519561).

A neighbor joining phylogenetic tree was constructed based on 16S rRNA sequences of strain YL2 and other published representative strains of 19 species in genus *Pseudomonas*, with *Thiopseudomonas* as an outgroup. The tree was generated by bootstrapping with 1000 replications [27]. Bar corresponds to an evolutionary distance of 0.005.

The image of strain YL2 were captured by a scanning electron microscopy, SEM (Zeiss, Oberkochen, Germany).

### 2.4. Construction of a SMX-Degrading Consortium

The resting cells were prepared by a method described previously [39]. Cells were harvested by centrifugation when cell growth reached mid-exponential phase. Harvested cells were washed with 0.02 M phosphate buffer solution (PBS, pH = 7) three times, and then resuspended in 0.02 M PBS (pH = 7) to an OD_660_ ≈ 1.0. Then, the resting cells were finally prepared by strict washing and centrifugation in PBS triples to ensure that no addition of any carbon source was introduced. The consortium was made up of an equal number of YL1 and YL2 resting cells.

### 2.5. Determination of Optimal Conditions

Four different factors involved in the biodegradation of SMX were tested: temperature, pH, shaking speed, and initial SMX concentration. Experiments were carried out in 250 mL Erlenmeyer flasks, filled with 100 mL of MSM containing 100 mg·L^−1^ of SMX. In addition, the study of optimal growth temperature was performed by setting five different temperatures at 10 °C, 20 °C, 30 °C, 40 °C, and 50 °C at 160 r·min^−1^. The pH of the media was adjusted to 5.0, 6.0, 7.0, and 8.0 with 1 M HCl and 1 M NaOH before inoculation to determine the optimum pH value. The reaction was initiated in a rotary shaker at 120, 140, 160, and 180 r·min^−1^. Moreover, the initial concentration of SMX was as follows: 50, 100, 200, 300, 400, and 500 mg L^−1^ to elucidate clearly the SMX degradation process. The experiments were carried out in triplicate.

### 2.6. Analysis Methods of Biotransformation

The removal of contaminants was assessed by measuring the TOC concentration in the supernatant. The TOC values of filtered samples were analyzed with an Analytikjena Multi N/S 2100S TOC analyzer.

Moreover, the cell growth curves were constructed based on the results obtained by spectrophotometry at 660 nm (OD_660_). The control group was cultured with an equivalent volume of PBS, adding inactivated resting cells. The OD_660_ of the experimental groups were subtracted from those of the control groups to remove the effect of abiotic factors [40]. The SMX detection method was based on Yu et al. [25]. The concentration of intermediate products was monitored by ultra performance liquid chromatography (UPLC). The residue sample was diluted to a range of 1–10 mg·L^−1^ as needed and determined by a UPLC (Waters Co, Milford, MA, USA) equipped with a photodiode array (PDA) detector. A 265-nm wavelength was set with a C18 column (2.1 × 50 mm, 1.7 μm). The mobile phase consisted of acetonitrile and purified water with 0.1% formic acid (40:60, *v*/*v*) at a flow rate of 0.3 mL·min^−1^. The column temperature was 30 °C, and the injection volume was 5 µL [41].

### 2.7. Quantitative PCR (qPCR) Analysis

The population density of each strain was monitored using qPCR targeting the gene encoding 16S rRNA. Table 1 shows the primer sequences. The qPCR protocol had a pre-denaturation at 95 °C for 5 min; then the temperature was kept at 95 °C for 15 s after denaturation, and a final annealing step at 60 °C for 1 min. In addition, the fluorescence was collected for 45 cycles. Standard curves were prepared using serial 10-fold dilutions of the near-complete fragment of the 16S rRNA gene of each strain. Gene copy numbers were calculated by the standard curve method as described elsewhere [42].

### 2.8. Whole Genome Sequencing and Assembly

After an overnight incubation at 30 °C with shaking at 160 r·min^−1^, cells of strain YL2 were harvested from the LB broth. The DNA genome was extracted by using the Genomic DNA Purification Kit, and the concentration of DNA sample was detected using a fluorometer or microplate reader. The genome of strain YL2 was sequenced using the Illumina HiSeq 2000 and Miseq platforms [43]. A total of 1.28 Gb high quality data (8,764,210 reads) were assembled with the SOAP de novo assembler (v2.04). The genome sequence was submitted to the GenBank database under accession numbers JAJSLU000000000.

### 2.9. Genome Annotation

The annotation of the YL2 genome was carried out by Rapid Annotations using the Subsystems Technology (RAST, Version 2.0) web service [44]. The SEED was used for predicting gene functions and discovering new pathways. NCBI Open Reading Frame Finder (ORF Finder) was used to predict the ORFs and they were annotated from non-redundant (NR), Swiss-Prot, and the Kyoto Encyclopedia of Genes and Genomes (KEGG) databases using the Basic Local Alignment Search Tool (BLAST). The predicted ORFs were mapped to KEGG database using the BLAST with a cut-off of 10^−5^ for the e-value to find the potential metabolic pathways [30]. The Comprehensive Antibiotic Resistance Database (CARD) analysis was carried out. The genome was screened with the Resistance Gene Identifier (RGI) for perfect and strict hits to known [45].

## 3. Results

### 3.1. Isolation and Identification Characterization of Strain YL2

During the isolation and purification process of YL1, a new strain was isolated using LB agar media and named YL2. The colonies of strain YL2 were circular, mucoid, and white-yellow after 2 days when cultured on LB agar. The isolated bacteria were Gram-negative and non-spore-forming rods of approximately 1 × 2 μm in size (Figure 1). Strain YL2 and strain YL1 could coexist when cultured on the SMX-containing MSM-agar plate. However, YL2 could not grow on MSM medium with SMX as the sole carbon source. As shown in Figure 1a, the colonies of YL2 could only be observed to be closed to those of YL1 on the SMX-MSM plates.

### 3.2. SMX Biodegradation Efficiency of the Consortium

A consortium consisting of an SMX-degrading strain (YL1) and a co-metabolic strain (YL2) was constructed, which was able to completely degrade SMX at an initial concentration of 100 mg/L within 48 h. In contrast, the degradation of 100 mg/L SMX by strain YL1 required 57 h, which was 6 h slower than the consortium. As in Figure 2, the TOC removal rate of the dual bacteria system was 38.94 ± 3.44%, while it was only 29.39 ± 2.58% in the single bacterial system.

In addition, independent sample T-tests were performed for the OD_660_, SMX concentration, and TOC concentration data (*p*-values were 0.105, 0.702, and 0.049, respectively). The presence of YL2 significantly affects the removal efficiency of TOC during the SMX degradation.

### 3.3. Effect of Different Growth Conditions on SMX Biodegradation by the Consortium

To examine the effect of different culture factors on the degradation efficiency of consortium, the temperature, shaking speed, pH, and initial SMX concentration were assessed. The optimal growth conditions for the consortium were 30 °C, pH = 7.0, a shaking speed of 160 r·min^−1^, and an initial SMX concentration of 200 mg·L^−1^ (Figure 3).

The consortium grew well on SMX at 20–40 °C, and the removal rate of SMX reached the maximum value at 30 °C at an SMX concentration of 100 mg·L^−1^ and pH 7, as shown in Figure 3a. When the temperature increased to 50 °C, no reduction in SMX concentrations was observed. At the same time, a low temperature was not suitable for consortium either. The optimum shaking speed of the consortium was 160 r·min^−1^ with the addition of YL2. As shown in Figure 3c, the optimum pH for the SMX biodegradation was assessed at 5, 6, 7, and 8. The degradation performance of the consortium was better in acidic conditions than in alkaline conditions. The highest degradation efficiency of both treatments (consortium and only YL1) was achieved at pH 7. When the initial concentration of SMX was 200, the degradation percentage of the consortium was maintained at a high level (97.30%), and the removal of SMX was significantly enhanced compared to that of monoculture at the same time (*p* = 0.001).

### 3.4. The Analysis of Composition in the Consortium during the Degradation of SMX

The qPCR of strain YL1 or strain YL2 revealed that these organisms grew simultaneously in this medium, maintaining a stable relative proportion throughout all the phases of growth. As the reaction proceeded, the absolute advantage of YL1 diminished compared with YL2 in the consortium. Eventually, the copy numbers between YL1 and YL2 were maintained close to a 1:1 ratio (Figure 4).

The overall trend of YL1 and YL2 copies in the consortium was consistent when the initial concentrations of SMX were different. The results of the qPCR showed that both the copies of YL1 only and the consortium increased first and then decreased with the increase of the SMX concentration. The maximum copies of YL1 in the consortium appeared at 200 mg·L^−1^, while the copies of YL1 reached the maximum value at the concentration of 100 mg·L^−1^. The copies of YL2 in the consortium were more than that of YL1 when the concentration was within 100 mg·L^−1^. When the substrate concentration exceeded 200 mg·L^−1^, the copies of YL1 were dominant. The difference in the expression reached the maximum value at 200 mg·L^−1^ and the copies of YL1 in the consortium reached 59.49%.

### 3.5. Utilization of the Major Intermediates by YL2 during the Degradation of SMX

The results showed that YL2 can utilize 4-aminophenol, but it cannot degrade and utilize 3A5MI. The growth was measured by an increase in OD_660_ of the liquid culture. Figure 5 shows that when the biomass increased, the concentration of 4-aminophenol in the culture began to decrease. The degradation of 4-aminophenol and the growth of YL2 occurred simultaneously, which implies that YL2 could grow on 4-aminophenol, while no significant growth of YL2 was achieved in the 3-amino-5-methylisoxazole treatment. These results confirmed that 4-AP could support the growth of strain YL2 as a sole carbon source.

### 3.6. Genomic Features and Putative Functional Classification

The genome of strain YL2 was found to have a size of 6,190,966 bp with a 61.20% G + C content. N50 and N90 values were 249,338 bp and 112,725 bp, respectively. The draft genome annotations and functional characterization by RAST allowed the identification of 5702 protein coding genes, 3 rRNA genes and 60 tRNAs, and 63 aromatic compounds genes. Five resistance genes in strain YL2 were identified by querying the Comprehensive Antibiotic Resistance Database (Appendix A).

The 4-aminophenol metabolism of YL2 was considered because it could provide a source of carbon which is essential for the growth of strain YL2, and the annotation of functional enzymes helped to identify the possible structure of the metabolic intermediates. The genetic characterization of the 4-AP transforming gene cluster from strain YL2 showed the presence of genes encoding the *p*-benzoquinone reductase (PnpB, 44.55% identity at the amino acid level), maleylacetate reductase (with 27.47% identity), 3-oxoadipate CoA-transferase (PcaI, 49.77%), and 3-oxoadipyl-CoA thiolase (PcaF, with 67.41% identity). Therefore, it can be proposed a probable composite degradation pathway (Figure 6). The 4-AP was deaminated by a 4-aminophenol dehydrogenase, resulting in *p*-benzoquinone, which was catalyzed by the *p*-benzoquinone reductase enzyme to produce hydroquinone. The hydroquinone was further transformed into 1,2,4-trihydroxybenzene, which was transformed to maleylacetate, and eventually entered the citrate cycle.

The genome annotation revealed the *ssuEADCB* gene cluster which encodes proteins for the utilization of sulfur from a wide range of aliphatic sulfonates (Figure 6). Blast results showed that the protein sequences of *ssuEADCB* had high similarity with FMNH_2_-dependent alkanesulfonate monooxygenase (100%) [46]. Sequence similarity searches indicated that the proteins encoded by *ssuA*, *ssuB*, and *ssuC* are likely to constitute an ABC type transport system, whereas *ssuD* and *ssuE* encode an FMNH_2_-dependent monooxygenase and an NAD(P)H- dependent FMN reductase, respectively [47]. In addition, according to the KEGG Database, strain YL2 carries complete biosynthesis pathways for riboflavin, which is the precursor of FMN (Figure 6).

## 4. Discussion

The separation of YL2 provided a reference for the construction of a stable co-metabolizing consortium for the degradation of SMX. Strain YL2 could form clear colonies on SMX-MSM plates until it was co-inoculated with strain YL1 and the incubation time was extended more than 7 days. Extended incubation time could adjust the survival mode of microorganisms in the environment with a low metabolic activity, which could help to isolate, culture, and identify them [48].

The consortium was shown to have the ability to promote the mineralization of SMX compared to strain YL1. To date, there is a shortage of knowledge about the degradation of SMX by co-culture. According to previous studies, co-culture of *Microbacterium* sp. BR1 and *Rhodococcus* sp. BR2 showed higher mineralization SMX degradation rates and shorter lag times compared to cultures with single strains. Both strain BR1 and BR2 were demonstrated to have the capacity to mineralize SMX [49]. Similarly, Reis et al. described a low abundance and slow-growing *Leucobacter* GP, which could degrade SMX, thriving only in co-culture with *Achromobacter denitrificans* PR1 [28,32]. It could effectively alleviate the instability of PR1 during the degradation of SMX and improve the ability of mineralization due to the degradation ability of strain GP itself. Contrary to previous studies, strain YL2 is a non-degrader but can play a supporting role in removing SMX. Recently, Zhao et al. found that the SMX biodegradation efficiency was enhanced by co-culture of *Shewanella oneidensis* MR-1 and *Paracoccus denitrificans* under anaerobic conditions. Strain MR-1 promoted the degradation of SMX from 48.9% to 94.2% within 7 d with glucose as the carbon source by enhancing NADH formation, electron transfer, and consumption [50]. This is the first time that a study has found that Gram-negative bacteria cannot degrade and effectively promote the performance of other degraders when SMX was used as a single carbon source. In terms of both the degradation capability of SMX and the growth ability of the bacteria, there was no significant difference between the consortium and the treatment with YL1 alone. YL2 cannot directly cleave the SMX compound, so it had a limited effect on the initial degradation of SMX. The copy number of YL2 was changed synchronously with that of YL1 without significant differences during the degradation process. However, there was a significant difference in the TOC removal.

The role of YL2 in the degradation might be related to the rapid utilization of degradation intermediates. The degradation of SMX by the consortium, as well as the removal of toxic transformation products such as 3-amino-5-methylisoxazole and 4-aminophenol showed that strain YL2 could grow on 4-AP. It had been reported by a number of researchers that *Pseudomonas* could use multiple aromatic compounds under aerobic conditions. *Pseudomonas* sp. strain ST-4 could utilize 4-AP in the presence of glucose without information on the accumulation of intermediates during 4-aminophenol degradation [51]. *Pseudomonas* strain AP-3 could grow on 2-aminophenol [52]. In addition, aniline and phenol could also be used as a source of carbon and energy for *Pseudomonas* [51,53,54]. Based on the genome sequence analysis, we proposed that strain YL2 degraded 4-AP via a typical hydroquinone pathway, as shown in Figure 6. Previously, *Pseudomonas* sp. strain WBC-3 utilizes *p*-nitrophenol as a sole source of carbon was reported and the roles of two quinone reductases in the microbial degradation of substituted aromatic compounds were illustrated [55]. A flavin monooxygenase (PnpA) is capable of converting *p*-nitrophenol to *p*-benzoquinone in the presence of NADPH. Likewise, strain YL2 could possibly require monooxygenase activity in the initial step of 4-AP metabolism. However, no comparable DNA sequences are currently available in the database. PnpB is a flavin mononucleotide and NADPH dependent *p*-benzoquinone reductase that catalyzes the reduction of *p*-benzoquinone to hydroquinone. Identification of PnpB and genes responsible for hydroquinone degradation in strain YL2 reveals its metabolic pathway for the degradation of 4-AP [56].

The presence of strain YL2 enhances the SMX degradation by adding riboflavin and enhancing the reducing capacity according to the genome of strain YL2. Riboflavin, an extracellular electron shuttles, is essential for normal cellular functions, which has a significant promotion effect on the transformation of SMX [57,58]. The ability to secrete riboflavin during aerobic growth seems to be a widespread phenomenon in *Pseudomonas*. It has been demonstrated that the metabolic expense of riboflavin secreting by *Pseudomonas* may be an energetically beneficial process, as it can be used in multiple rounds of extracellular electron shuttling in this local environment [59]. Comparing the riboflavin metabolism pathways of strains YL1 and YL2 obtained from the KEGG database, two enzymes with the same function but different enzyme classification were highlighted. The FMN reductase (EC 1.5.1.36) in strain YL1 encoded by *sadC* plays an important role in the degradation of SMX. It is responsible for the initial attack on the sulfonamide molecule together with the monooxygenase encoded by *sadA* and functions together with another monooxygenase encoded by *sadB* to convert 4-AP to 1,2,4-trihydroxybenzene [25]. The FMN reductase (EC 1.5.1.38) from strain YL2 catalyzes the reduction of free flavins by NADPH, in contrast with EC 1.5.1.36, which uses NADH as an acceptor. The reducing ability can be responded to by both NADH/NAD+ and NADPH/NADP+. It has been shown that an increase in reducing capacity can promote the degradation ability of the synthetic colony to SMX. As a result, one of the reasons to promote SMX mineralization is the improvement of the reducing capacity as a result of the participation of YL2. Genomic information of consortium members helps us to understand interspecific interactions better and suggest possible 4-AP metabolic pathways.

## 5. Conclusions

The biodegradation of SMX by a consortium consisting of a metabolic strain YL1 and a co-metabolic strain YL2 were assessed. The symbiotic effect of the consortium slightly increased the biodegradation speed of SMX, but the TOC removal was 38.94% which was significantly higher than that of the monoculture system. In addition, during the degradation of SMX, the biomass of YL1 and YL2 increased synergistically implying that the constructed consortium was able to maintain a stable community structure. Furthermore, strain YL2 could remove 19.56% of 4-AP, one of the intermediates produced when metabolizing SMX by strain YL1, within 5 days. Moreover, genomic information of consortium members helps us to understand interspecific interactions better and suggest possible 4-AP metabolic pathways. These results contribute to the understanding of the metabolism of bacterial communities, which is essential for the bioremediation of antibiotic-contaminated environments.

## Figures and Tables

**Figure 1 microorganisms-10-00648-f001:**
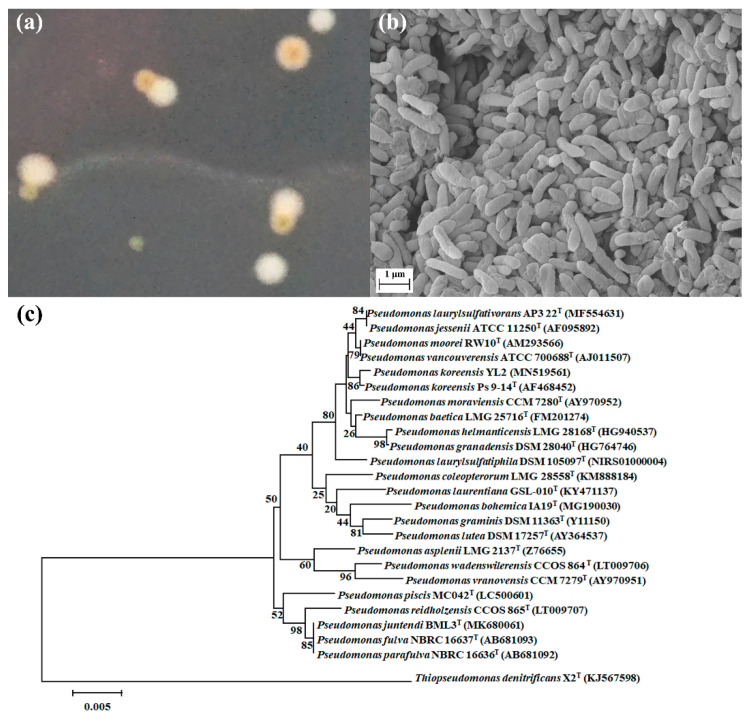
Colonies on MSM–SMX–agar plates. Orange colonies of the strain YL2 grew around the beige colonies of strain YL1 (**a**); Scanning electron microscopy (SEM) image of strain YL2 (**b**). Phylogenetic tree showing the relationship between strain YL2 and other related strains based on 16S rRNA gene sequences (**c**).

**Figure 2 microorganisms-10-00648-f002:**
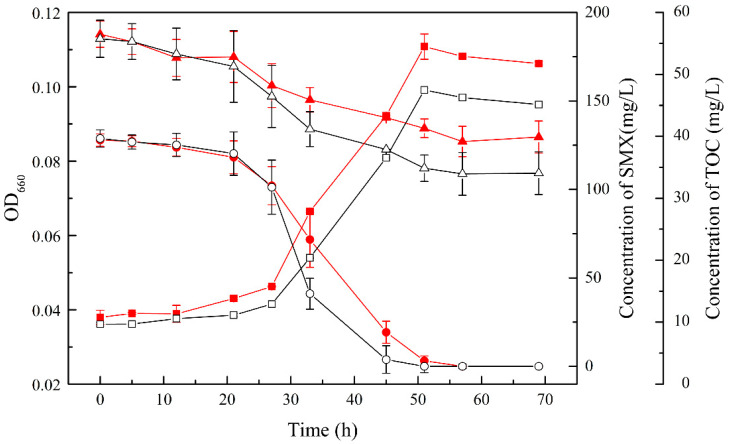
Removal of SMX during the growth of the strain YL1 (red) and the consortium (black). The circle shows the concentration of SMX; the triangle represents the concentration of TOC; square represents OD_660_.

**Figure 3 microorganisms-10-00648-f003:**
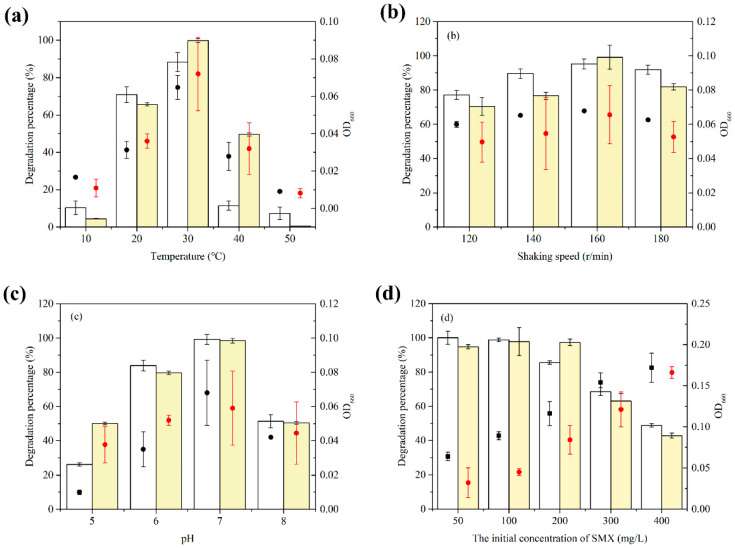
Effects of different conditions on the growth and degradation of the consortium with SMX as a sole carbon source. Columns represent the degradation percentage of SMX. (**a**) Effects of incubation temperature. (**b**) Effects of the rotation speed of shaking. (**c**) Effects of media pH. (**d**) Effects of the initial concentration of SMX.

**Figure 4 microorganisms-10-00648-f004:**
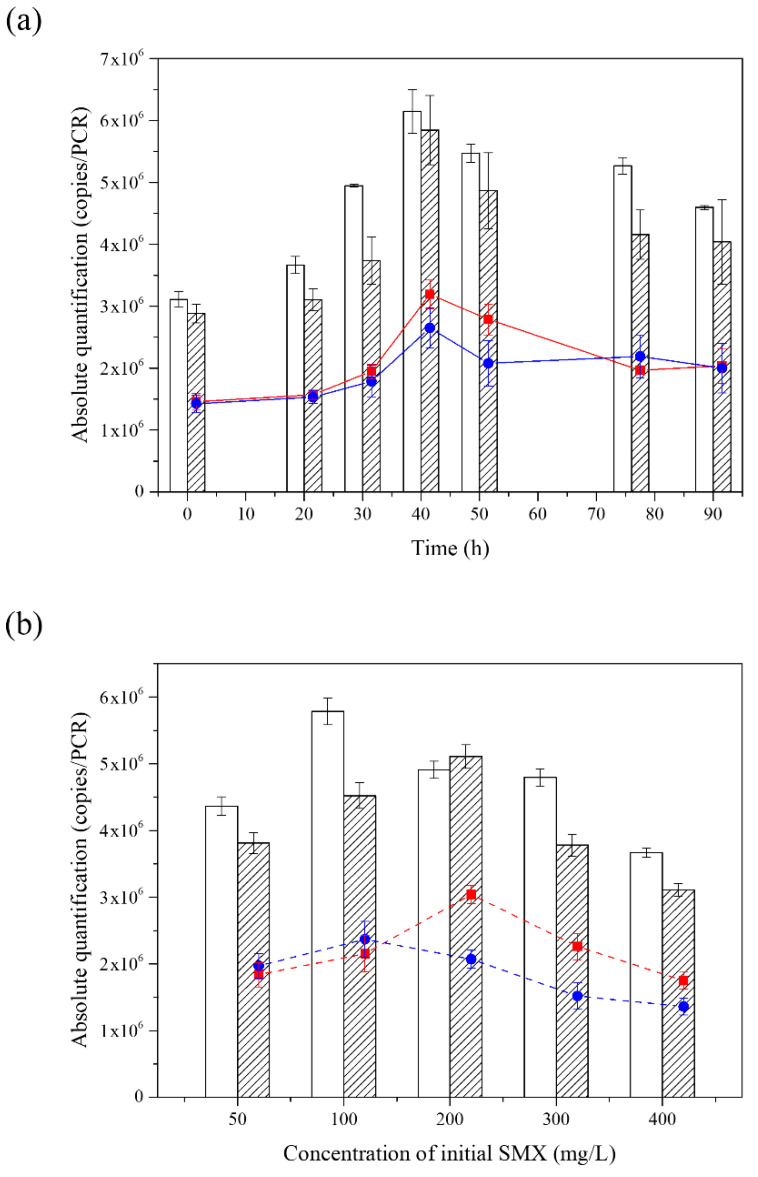
The bar chart represents the consortium (diagonal lines) and the strain YL1 only (white bar). The red line represents the copies of YL1 in the consortium, and the blue line represents the copies of YL2 in the consortium. (**a**) Abundance of YL1 and YL2 during degradation of SMX at 200 mg·L^−1^. (**b**) Abundance of YL1 and YL2 at different initial SMX concentrations.

**Figure 5 microorganisms-10-00648-f005:**
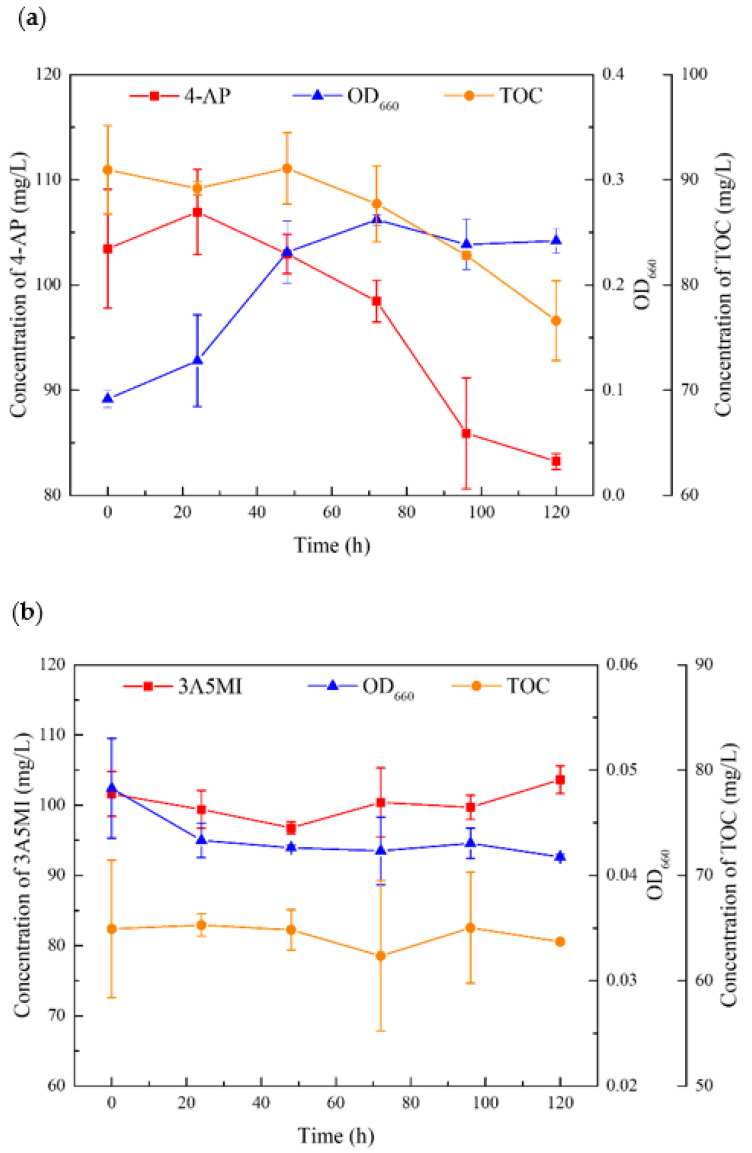
The ability of YL2 to utilize SMX degradation products: (**a**) Growth of strain YL2 on 4-aminophenol and the corresponding biodegradation with time; (**b**) Growth of strain YL2 on 3-amino-5-methylisoxazole and the corresponding biodegradation with time.

**Figure 6 microorganisms-10-00648-f006:**
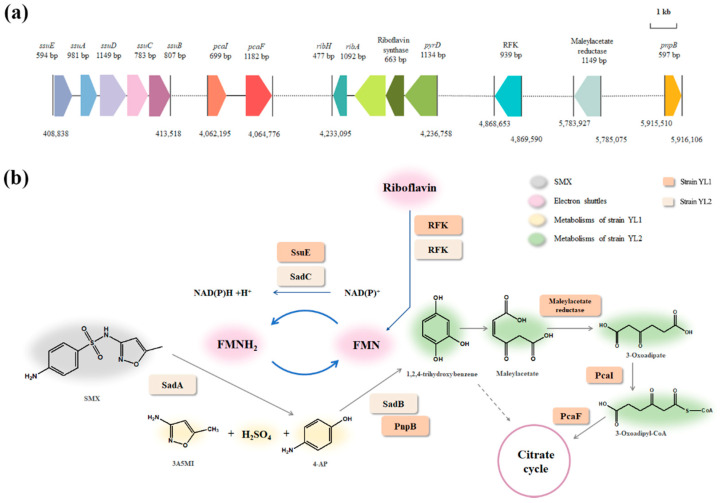
Genes associated with the promotion of SMX degradation in strain YL2 and their localization in the genome (**a**). Proposed pathway of SMX metabolism in the consortium relying on FMN as an electron shuttle. The dashed line indicates the open-loop pathway, which was not found in strain YL1 (**b**).

**Table 1 microorganisms-10-00648-t001:** The names and sequences of quantitative primers.

Target Genes	The Names of Quantitative Primers	The Sequences of Quantitative Primers
16S rRNA gene of strain YL1	YL1 16S rRNA-F	TCCTCAGCGTCAGTTACA
	YL1 16S rRNA-R	TGGTGTAGCGGTGAAATG
	YL1 16S rRNA-probe	FAM-AGAGACCTGCCTTCGCCATCGG-MGB
16S rRNA gene of strain YL2	YL2 16S rRNA-F	ATGCGTAGATATAGGAAGGAA
	YL2 16S rRNA-R	CAGGCGGTCAACTTAATG
	YL2 16S rRNA-probe	FAM-ACCACCTGGACTGATACTGACACTGA-MGB

## Data Availability

Publicly available datasets were analyzed in this study. These data can be found at https://www.ncbi.nlm.nih.gov/ (accessed on 28 February 2022) in GenBank, accession numbers JAJSLU000000000 and MN519561.

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
