# Peer review of "Harnessing Paenarthrobacter ureafaciens YL1 and Pseudomonas koreensis YL2 Interactions to Improve Degradation of Sulfamethoxazole"

_microorganisms, 2022, doi:10.3390/microorganisms10030648_

Round 1

Reviewer 1 Report

 I reviewed the manuscript entitled, Harnessing Paenarthrobacter ureafaciens YL1 and Pseudomonas koreensis YL2 interactions to improve degradation of sulfamethoxazole. In my opinion, the manuscript is well written with appropriate scientific literature. Introduction and methodologies are described with detailed information. Conclusions are supported by the research findings. In my opinion, the manuscript can be accepted for publication after addressing the suggestions below.

 Abstract should be revised by focusing on experimental findings

Line 56: illustration… please use better word

Lines 64 to 67: not necessary here

Line 74: Kermel Chemical Reagent Ltd…city and country?

Methodologies are appropriate with related citations

Figure 3. statistical analysis must be performed to understand the differences

Figures quality must be improved

Check the references and format according to journal guidelines. 

Reviewer 2 Report

Several studies suggest that the traditional activated sludge process in the sewage treatment plants is not designed for the efficient removal of emerging pollutants such as sulphonamide. The research manuscript describes the isolation and characterization of a microbial consortium for improvement of SXT degradation. The authors used several methodologies to uncover the mechanism of SMX degradation by the consortium. The results obtained could contribute to the development of functional microbial communities for bioremediation of antibiotic-contaminated environments.

Author Response

We appreciate for Reviewers’ warm work earnestly.

Reviewer 3 Report

I suggest that the images of figures 1, 2 and 5 be sent with higher quality
